# Isokinetic Dynamometry as a Tool to Predict Shoulder Injury in an Overhead Athlete Population: A Systematic Review

**DOI:** 10.3390/sports8090124

**Published:** 2020-09-08

**Authors:** Andrea Bagordo, Kimberly Ciletti, Kevin Kemp-Smith, Vini Simas, Mike Climstein, James Furness

**Affiliations:** 1Water Based Research Unit-Bond Institute of Health and Sport, Bond University, Gold Coast, QLD 4226, Australia; andrea.bagordo@student.bond.edu.au (A.B.); kimberlyrose.ciletti@student.bond.edu.au (K.C.); kkempsmi@bond.edu.au (K.K.-S.); vsimas@bond.edu.au (V.S.); michael.climstein@scu.edu.au (M.C.); 2Clinical Exercise Physiology, School of Health and Human Sciences, Southern Cross University, Bilinga, QLD 4225, Australia; 3Physical Activity, Lifestyle, Ageing and Wellbeing Faculty Research Group, University of Sydney, Sydney, NSW 2006, Australia

**Keywords:** isokinetic dynamometry, shoulder, internal and external rotation, sport, injury, prevention, systematic review

## Abstract

Prospective and cross-sectional studies have used pre-season isokinetic dynamometry strength and endurance measurements of shoulder internal rotation (IR) and external rotation (ER) to determine if they can be correlated to injury. However, to date, no review has provided a synthesis of all available literature on this topic. The aim of this systematic review was to identify isokinetic dynamometry studies that assess shoulder IR and ER strength and endurance in the overhead athletic population in relation to shoulder injury. Electronic databases (PubMed, CINAHL, and SportDiscus) were searched through September 2019 using pre-determined search terms. Both prospective and cross-sectional studies were included in this review. Studies were assessed for quality using either Appraisal Tool for Cross-sectional Studies (AXIS) or Critical Appraisal Skills Programme (CASP). Data on outcome measures of strength and endurance peak torque (PT) and ratios (ER:IR) were extracted and further analysed using a best evidence synthesis approach. A total of 13 articles met the inclusion criteria. Conflicting evidence was found when reviewing all studies without differentiating by study type. Prospective study designs revealed strong evidence that reduced IR endurance and reduced strength ratios are predictive of shoulder injury. Cross-sectional literature showed only conflicting and limited evidence for all outcome measures. At this stage, more research is needed in individual sporting populations using prospective cohort designs.

## 1. Introduction

Isokinetic dynamometry has become an increasingly popular assessment tool used in exercise science and sports medicine. An isokinetic dynamometer measures the applied force throughout a specified range of motion and provides information about dynamic concentric and/or eccentric muscle contractions at a specified speed [1]. Isokinetic dynamometry is widely recognised as the gold standard for measuring muscle strength and muscle endurance through a specified range of movement [2]. Current literature has identified that isokinetic dynamometry has been used in a variety of settings, which includes injury prediction, research, and rehabilitation [3,4].

In the sporting population, isokinetic dynamometry has many uses, which includes: (1) as a pre-season measure to determine baselines for return to play criteria [4] and measuring muscle imbalances to predict injuries in the lower limb [3], (2) as a profiling tool for athletes within their sport in relation to gender, age, or body mass [5,6,7], (3) as a comparison tool to differentiate between levels of sport [8], playing position [9], and pre-season and post-season strength measurements [10], (4) as part of a return to sport criteria following anterior cruciate ligament reconstruction [11,12] or hamstring injuries [13], and (5) to determine strength imbalances between muscle groups as a method to improve performance [14]. The focus of the previously mentioned studies has been on the lower limb, revealing a large gap in the literature for the upper limb, specifically the shoulder.

The evaluation of the shoulder is often used to assess functional stability, dynamics, and muscular performance (strength, speed, and endurance) in sports that are predominantly overhead. Additionally, isokinetic dynamometry evaluation has also been used to determine profiles or athletes and patients who demonstrate shoulder abnormalities. This evaluation has been used in clinical decision-making and rehabilitation [15,16,17]. 

Shoulder injuries are a major concern with overhead sports such as volleyball, baseball, handball, rugby, swimming, and surfing due to their high rate of reported injury. The incidence of shoulder injuries in the overhead sporting population has been described in the literature as 0.2/1000 h and 1.8/1000 h [18,19,20]. Furthermore, overuse injuries at the shoulder have been reported to be 10% and 37%, respectively [18,19]. In one swimming cohort, shoulders were found to be the most common site of injury accounting for 38% of major injuries [21]. Furthermore, a study conducted by Furness et al. [22] found the main, acute injury-prone location in recreational and competitive surfers was the shoulder (16.4%). The typical movement pattern required in overhead sports consists of shoulder internal rotation (IR) and external rotation (ER). Given the primary movements used in throwing and swimming, the current study chose to focus on shoulder IR and ER since it most similarly replicates the actions and requirements seen in overhead sports and can be most accurately tested. With such high incidence rates of shoulder injuries, prevention is key to reduce the risk of injury in any athlete.

Injury prevention is an important aspect of modern sports medicine. Approximately 22% of athletes with shoulder injuries are out of play for more than three weeks [23]. Therefore, coaches, physiotherapists, athletic trainers, strength and conditioning specialists, and athletes themselves all value information on how they can best reduce the risk of injury and decrease time lost from training and competition. In addition, one study following major league baseball players found that only half of pitchers with pre-season shoulder injuries returned to compete in the same season and, of those, half experienced re-injury that prevented them from playing for the remainder of the season [24]. Isokinetic dynamometry has been used to help predict and prevent injuries as previously mentioned above. To date, there are inconsistencies in the literature on strength disorders in relation to soft tissue injury. Articles have found that strength imbalances assessed by isokinetic dynamometry are linked to injury [3] while others show that it is not a predictor [25]. Numerous articles provide recommendations on the rehabilitation of strength imbalances to prevent injury [26,27,28,29]. However, there is conflicting evidence to support that strength imbalances lead to shoulder injuries. A recent systematic review on risk factors and prevention of shoulder injuries in overhead sports showed that the evidence for prevention measures in this population is limited [30]. However, this review did not include the use of isokinetic dynamometry. 

Following a thorough and extensive review of the literature, there is currently no systematic review specific to strength measurements of shoulder IR and ER measured using isokinetic dynamometry IR, ER and IR, ER ratios and association with injury. Given the high rate of shoulder injury across a variety of overhead predominant sports, assessment of IR and ER via isokinetic dynamometry may provide valuable insight into injury prevention. Therefore, the aim of this study was to systematically review the literature and identify studies using isokinetic dynamometry to assess shoulder IR and ER strength and endurance in the overhead athletic population in relation to shoulder injury. 

## 2. Materials and Methods

### 2.1. Study Design

The study followed the methodology proposed in the Preferred Reporting Items for Systematic Reviews (PRISMA) statement [31]. In line with the PRISMA guidelines, a detailed search strategy was developed (Table 1), and a search was conducted and uploaded to the EndNote reference management software (EndNote X8.0.1, Clarivate Analytics, Boston, MA, USA) on 24 September 2019. A systematic literature review protocol was developed prospectively prior to data extraction and critical appraisal.

### 2.2. Search Strategy

A comprehensive, multi-step search strategy using PRISMA guidelines was conducted to identify relevant studies regardless of publication date. The databases searched were chosen based on their large number of peer-reviewed material in this area of interest and included: PubMed, SPORTDiscus, and CINAHL. The final search was designed with the aid of an experienced librarian at the Bond University Library, Gold Coast, Australia.

Search terms were identified by completing a rapid literature review, testing different key words, and discovering the common terms used in research relevant to this review. The search was conducted using search terms from five key subject areas: isokinetic dynamometer, shoulder, sport, injury, and prevention (Table 1). The Boolean Operators “OR” and “AND” were used to combine the search terms within and between each of the five subject areas, respectively. Researchers also conducted independent searches on Google Scholar using the question and statement, “Is isokinetic dynamometry predictive of shoulder injury in athletes?” and “Pre-season use of isokinetic dynamometry.” Search strategies did not need to be adjusted per database and no MeSH terms were included in the final search strategy.

### 2.3. Study Selection

Search results were imported into the EndNote reference management software where duplicate records were removed by variations of the title, author, and date. Titles and abstracts of retrieved articles were screened against predetermined eligibility criteria (details below). Any title and abstract that did not clearly investigate the shoulder strength or endurance of athletes using isokinetic dynamometry in relation to injury were discarded as being not relevant. After the initial screening, the full texts deemed eligible were retrieved for further analysis by two of the authors. The PRISMA diagram (Figure 1) outlines the search process in its entirety.

### 2.4. Eligibility Criteria

Eligibility criteria were discussed and agreed upon by all investigators prior to screening articles to best answer the research question “Is shoulder strength and endurance assessed by isokinetic dynamometry associated with shoulder injury in overhead sports?”. Studies were included if they were in English, published in a peer-reviewed journal, with the full-text available and no set limit on the date of publication. Human participants (male or female) of any age and playing level, from any overhead sport (overhead sport: sport in which an athlete must repetitively lift the arm above head), were eligible. Studies included needed to utilise isokinetic dynamometry as an intervention and be either observational design (prospective or retrospective cohort study or cross-sectional study) to ensure all existing literature meeting the aim would be reviewed. Additionally, studies needed to examine shoulder strength and/or endurance, include shoulder IR and ER as a measurement outcome, and either analyse the risk of injury (prospective or retrospective cohort) or compare uninjured versus uninjured participants (cross-sectional). Any study not meeting the inclusion criteria or that only profile strength values or injuries in a sporting population and did not link outcomes to injury were excluded.

To minimise bias, search terms were specific to the research question, duplicates were removed, and an inclusion/exclusion criterion were established prior to screening a limit of inclusion bias. To limit selector bias, two researchers independently screened and selected studies, any disagreements regarding which studies should move to the next stage were resolved by discussion and consensus.

Studies that met the inclusion criteria were retrieved in full and assessed in detail against the inclusion criteria. Full-text studies that did not meet the inclusion criteria were excluded and reasons for exclusion are provided in the PRISMA flow diagram (Figure 1). Full-text articles that met the inclusion criteria underwent a process of critical appraisal. The results of the search are presented in the PRISMA flow diagram (Figure 1). Any disagreements between the reviewers were resolved through discussion or by a third researcher. Through this approach, search bias, duplication bias, inclusion criteria bias, and selector bias were limited [32].

### 2.5. Critical Appraisal/Assessment of Methodological Quality

Critical appraisal was performed independently by two researchers. It was agreed that, for cohort studies, a modified version of Critical Appraisal Skills Programme (CASP) [33] would be used and a modified version of AXIS [34] would be used for cross-sectional studies. The modifications described below were not considered to have impacted the quality of study analysis. To ensure consistency, guidelines for each question were discussed and agreed upon prior to screening any articles. A pilot article was used for both CASP and AXIS to ensure there was a mutual understanding between the appraising researchers and the senior researcher prior to the appraisal.

CASP [33] is a 12-question appraisal tool used for prospective and retrospective cohort studies. Each question is answered with a ‘Yes,’ ‘No,’ or ‘Can’t tell.’ The CASP checklist does not have a formal scoring system. Therefore, a point system was created for the purpose of appraising the quality of the evidence. An answer of ‘Yes’ was awarded 1 point, and an answer of ‘No’ or ‘Can’t tell’ was awarded 0 points, with the maximum score being 12. Questions 7, 8, and 9 were combined since they were similar in appraising the quality of the results section.

The appraisal tool for cross-sectional studies AXIS [34] is a 20-question tool used to systematically assess and judge the reliability of studies by addressing issues often apparent in cross-sectional studies. Questions 7, 13, and 14 were removed from the questionnaire since they showed little relevance to the articles being reviewed due to not having a non-response bias. Question 19 was related to the conflict of interest and was modified and scored differently due to the nature of the question being awarded 0 points for an answer of ‘Yes’ and 1 point for an answer of ‘No.’ Therefore, a point was awarded for having nil conflicts of interest. Question 20 was modified to include that there must be both ethics AND consent mentioned. Each question is answered with a ‘Yes,’ ‘No,’ or ‘Don’t Know.’ AXIS does not have a formal scoring system. Therefore, one was created to allow for appraising the quality of evidence. An answer of ‘Yes’ was awarded 1 point, and an answer of ‘No’ or ‘Don’t Know’ was awarded 0 points with the maximum score being 17. Modifications to critical appraisal tools such as the CASP and Axis whereby questions are assigned a point system have been previously used in other reviews [35,36].

After final review by the two researchers, each of the 13 articles were given a critical appraisal score (CAS) according to Kennelly [37] where each article was scored as a percentage and categorised as good (>60%), fair (45–59%), or poor (<45%), as seen in Table 2.

To determine the inter-rater reliability between the two scoring researchers, Cohen’s Kappa coefficient (κ) was calculated using SPSS software (IBM SPSS Statistics for Macintosh, Version 24.0. IBM, Armonk, NY, USA).

### 2.6. Data Extraction

Following the selection, critical appraisal and scoring of eligible studies, key data was extracted and tabulated by two researchers independently and then compared. Any discrepancies were reviewed, and a consensus was formed via discussion. Extracted data included the author(s), title, year, aim(s), study design, level of evidence, details about participants, testing protocol, and key findings. Due to the lack of definitions for strength and endurance associated with isokinetic dynamometry in the included literature, researchers discussed and agreed upon criteria for each. Strength was defined as any isokinetic dynamometer measurement <240°/s and <20 repetitions. Conversely, endurance was defined as any measurement ≥240°/s OR ≥20 repetitions OR defined as “fatigability.” The same criteria were applied to peak torque (PT) ratios to differentiate between strength and endurance ratios.

The level of evidence provided by each included study was extracted and graded according to the Australian National Health and Medical Research Council (NHMRC) criteria, as seen in Table 3 [38]. This criterion ranges from level I evidence, the highest available level, which includes systematic reviews of all randomized controlled trials to level IV evidence, which include case series [38,39].

### 2.7. Data Synthesis/Analysis

A critical narrative synthesis of key findings from the included studies was then conducted according to the aims of this systematic review. Researchers created a data extraction table that included each article and all relevant information pertaining to the aims of this review, which was then divided into two tables (Tables 3 and 4). Tables were cross-checked for accuracy by two separate researchers. When analysing the findings from each included study, researchers considered the methodological quality including both the CAS quality rating and the NHMRC criteria.

Meta-analysis was not possible in this study due to the heterogeneity of the included studies including the differing outcomes assessed in the included studies. Therefore, a best evidence synthesis approach was utilised to provide a qualitative analysis of the data (Table 5) using the five levels of evidence [40,41], which states:Strong evidence: provided two or more studies with high quality and by generally consistent findings in all studies (≥75% of the studies reported consistent findings).Moderate evidence: provided by one study with high quality and/or two or more studies with low quality, and by generally consistent findings in all studies (≥75% of the studies reported consistent findings).Limited evidence: only one study with low quality.Conflicting evidence: inconsistent findings in multiple studies (<75% of the studies reported consistent findings).No evidence: when no studies could be found.

### 2.8. Assessment of Sample Size

The sample sizes of included studies were further analysed by researchers using Hsieh [42] (‘Sample Size Tables for Logistic Regression’). This ensured valid statements were made on the overall findings included in this systematic review. Studies with a small sample size that detected a statistically significant relationship were kept in the final analysis of the best evidence synthesis (Table 5). However, articles that did not detect a statistically significant relationship and had a sample size that did not meet requirements laid out by Hsieh et al. [42] were removed due to a sample size that is too small to truly detect differences between groups (Table 6).

## 3. Results

### 3.1. Search Results

The search results are illustrated in the PRISMA flow diagram (Figure 1). The final search resulted in 537 articles, followed by removal of 173 duplicates, which left 364 articles to screen. These articles were screened by title and abstract against the predetermined inclusion criteria. Following the screening, the full text of 41 articles were retrieved and thoroughly assessed for inclusion in this systematic review. Of these, 13 articles (11 from the three online databases used and three hand-searched) met the inclusion criteria and were included in this systematic review.

### 3.2. Critical Appraisal Results

The CAS scores from both reviewers were compared, and Cohen’s Kappa coefficient (κ) analysis was conducted to determine the level of agreement. Kappa analysis revealed substantial agreement of the scoring results between raters in using the CASP tool and almost perfect agreement using the AXIS tool for critical appraisal. The overall agreement with CASP was (κ = 0.77) and AXIS was (κ = 0.81). Following the Kappa analysis, any discrepancies in scores were discussed and resolved by finalising the critical appraisal scores. The results of the CAS for included studies using the modified AXIS and modified CASP are shown in Table 2.

The common weaknesses identified within the articles during the critical appraisal process using the AXIS were that none of the included cross-sectional articles justified their sample size. When using the CASP tool, the prospective cohort studies were weak in identifying and accounting for confounding factors in their design and analysis.

### 3.3. Key Findings

#### 3.3.1. Characteristics of Included Studies

Two researchers independently reviewed all 13 articles and a detailed description of the included studies are provided in Table 3**.** Of the included studies, six were prospective cohort studies [43,44,45,46,47,48] and seven were cross-sectional studies [26,27,28,29,43,44,45].

#### 3.3.2. Participants

A total of 439 participants were included in the 13 studies with sample sizes ranging from a minimum of 14 to a maximum of 108 participants [26,27,28,29,43,44,45,46,47,48,49,50,51]. Six studies tested only males [26,27,46,49,51], four studies tested only females [26,28,29,43,50], and three studies tested both males and females [44,45,47]. Studies included participants aged between 10–30 years [26,27,28,29,43,44,45,46,47,48,49,50,51]. Most studies included participants from a single sport. However, one study combined handball and volleyball players [43]. The sport most identified was volleyball (four articles total, including the one combined sport study) [28,43,47,51]. Three studies investigated handball players (including the combined sport study) [43,46,50], two studies had subjects who were cricket fast bowlers [26,29], two studies used baseball players [27,49], two studies used swimmers [44,45], and one study used rugby league players [48]. Two studies compared athletes to healthy non-athletes [27,50].

#### 3.3.3. Testing Protocol

Eight studies implemented a seated position for isokinetic testing [27,28,29,43,44,48,49,50]. Three studies used a supine position [46,47,51], one study used a prone position [45], and one study did not specify the testing position [26]. The included studies did not specify the angle of the back rest, the seat pan, or any rotatory setting of the seat that may have been utilized. Seven studies included information about the range of motion allowed through the position using soft stops during testing with the majority having between 50° of IR to 50° of ER [26,27,43,46,47,50,51]. All but one study [48] included information about the resting period, which ranged from 10 s to 5 min [26,27,28,29,43,44,45,46,47,49,50,51]. Studies used a variation of testing speeds per second including 30°/s, 60°/s, 90°/s, 120°/s, 150°/s, 180°/s, 240°/s, and 300°/s, with 60°/s being the most common [26,27,28,29,43,44,45,46,47,48,49,50,51].

A follow-up of participants in the six prospective cohort studies [46,47,48,49,50,51] ranged from a weekly in-season questionnaire to monthly in-season questionnaires. No studies followed up the participants beyond a single sporting season.

### 3.4. Main Findings in Relation to Isokinetic Dynamometry and Its Association with Injury

The 13 studies reviewed identified four different outcomes: (1) strength (peak torque, PT), (2) endurance, (3) strength ratios (PT ratio), and (4) endurance ratios. Key findings for these outcomes can be found in Table 4**.** Results were compared for all 13 included studies (Table 5) and then further divided to separate the cross-sectional from prospective studies (Table 6).

When comparing outcomes from all 13 included articles regardless of type of study, the results showed inconsistent findings with overall conflicting evidence. After separating the two types of studies (prospective and cross-sectional) and excluding articles that did not meet an adequate participant size [42], clearer overall findings were revealed and outlined below.

#### 3.4.1. Cross-Sectional Studies

Seven studies in this review were cross-sectional [26,27,28,29,43,44,45]. Six of the cross-sectional studies looked at strength as an outcome measure [26,27,28,29,43,44] and two of these articles found that strength measurements were higher in the uninjured group [27,43]. One study found that strength was higher in the injured group [26]. Three studies discovered that there was not a significant difference between the strength (PT) of injured and uninjured participants [28,29,44]. These findings are inconsistent and received a rating of conflicting evidence, according to the best evidence synthesis criteria [38,39]. After discarding the articles that did not have significant findings and used small, non-justified sample sizes, three articles remained [26,27,43]. Of these, two good quality articles [27,43] found an association between a lack of strength and injury and one [26] found that injured participants were stronger. However, this article was rated as lower quality (fair). Due to the lack of high-quality studies, it was difficult to draw any firm conclusions between the lack of strength and injury. Two of the cross-sectional studies indicated that there may be some inference that strength variations may be a precursor to an injury. However, it is difficult to draw a formal conclusion due to the lack of rigor in the study design.

Only one cross-sectional study observed endurance as an outcome measure [43]. This article was rated as ‘good quality’ and found that endurance was higher in the uninjured group, which means injured participants had significantly lower endurance measurements. Therefore, in the cross-sectional literature, there is limited evidence for low endurance measurements shown in participants with shoulder pain or injury.

All seven cross-sectional articles observed strength ratios as an outcome measure [26,27,28,29,43,44,45]. Four articles found no significant difference between the strength ratios of injured and uninjured subjects [26,29,43,45]. After removal of articles that did not meet criteria for significance and sample size, three good quality articles remained [27,28,44]. Two of these articles agreed that strength ratios were higher in uninjured subjects [27,28] and one article found that strength ratios were higher in the injured subjects [44], which makes the results conflict.

Only one cross-sectional article noted endurance ratios as an outcome measure [45]. This study found that injured participants had lower endurance ratios (ER:IR). However, there is limited evidence for this as there was only one study. Overall, cross sectional articles only revealed limited and conflicting evidence with regard to whether isokinetic dynamometer measurements are associated with injury $\bar{x}$.

#### 3.4.2. Prospective Cohort Studies

Six studies in this review used a prospective cohort study design [46,47,48,49,50,51]. When looking at all of these articles, there was conflicting evidence for all outcome measures (Table 5). However, when disregarding articles who did not meet significant findings with a low sample size, clearer results were revealed.

Each of the six prospective studies reported on strength (PT) outcomes. Researchers compared PT results between the prospective studies and found that five of the six studies [46,48,49,50,51] showed no difference between strength values for the participants who became injured during the following season and those who did not, whereas only one high quality study [47] found an association between low PT and development of injury throughout the sporting season. This indicates that there is limited evidence for the association between PT and development of injury.

Four of the prospective studies [46,47,49,50] evaluated the ability of endurance to predict injury. Two articles showed no difference in measurements between the injured and uninjured [47,50]. These findings are inconsistent, which indicates conflicting evidence overall (Table 5). After discarding the evidence that does not meet the required number of participants for a valid outcome, two high quality studies remained [46,49]. Both articles are consistent in showing an association between low endurance measurements and development of injury during the sporting season. Therefore, there is a strong level of evidence for low endurance measurements as a predictor of injury. This was particularly evident in the measurement of the IR movement.

Strength ratios were measured in five of the prospective studies [46,47,48,50,51]. Three of the five articles agreed that PT ratio measurements were not different between athletes who became injured and the those who remained uninjured [46,47,48]. Two high quality articles [50,51] showed that athletes with lower strength ratios became injured. Therefore, there is strong evidence in this review that a low strength ratio is correlated with developing a shoulder injury.

Endurance ratios were only measured by four prospective articles [46,47,49,50] in which all had conflicting results. After removal of two articles [46,47] due to having a low number of participants, two high quality articles [49,50] remained, which showed that high endurance ratios were associated with injury. Due to the conflict in results, there are conflicting results for endurance ratios in this review.

Overall, the high-quality prospective literature has revealed strong evidence that athletes with reduced IR endurance and/or strength ratios develop a shoulder injury throughout the subsequent season (Table 6). Refer to Table 4 for details on these measurements.

Table 5 outlines the effects shown in each article for each of the outcomes measured, and their overall best evidence synthesis rating. It is an overarching picture of the research and does not differentiate between cross-sectional or prospective cohort studies nor does it consider effects that did not meet statistical significance, according to Hsieh’s [42] guidelines. A positive finding indicates that low measurements in that area were associated with injury. For ratios, a positive finding indicates that the non-injured group had a higher ratio.

Table 6 compares the results after separating cross-sectional and prospective cohort studies after the removal of non-significant findings in articles that did not reach the participant requirement outlined by Hsieh [42]. This simplistic view makes it easier to view the statistically significant evidence and the relationships between the studies.

## 4. Discussion

This was the first systematic review using isokinetic dynamometry measurements of IR and ER to predict shoulder injury in overhead athletes. Included in this review are 13 studies, six prospective studies, and seven cross-sectional studies. The findings of this study may help guide rehabilitation professionals working with overhead athletes on injury prevention methods for the shoulder. Best evidence synthesis of all articles revealed conflicting results. However, following removal of studies with low sample sizes and further pooling into the higher quality prospective studies only, strong evidence for isokinetic measurements of low endurance and low strength ratios as an injury predictor were shown.

Cross-sectional studies provide limited evidence since they only compare measurements between injured and uninjured subjects at a single point in time. This makes it unclear as to whether low outputs in the injured groups are due to having pain or injury resulting in apprehension during testing rather than having a genuine weakness. In contrast, prospective studies have a follow-up with the participants after a period to determine if having a low output led to the development of an injury at a later date.

In this review, the prospective research showed strong evidence that lower endurance measurements for IR and low strength ratios were indicative of developing a shoulder injury. A possible explanation for this finding is that the shoulder is the most mobile joint in the entire body and, consequently, lacks bone stability, which makes it a relatively weak joint compared to other ball-and-socket joints, such as the hip. The shoulder allows a full 360° range of motion, which is a necessary functional mobility in most overhead sports. The stability and integrity of the joint is reliant upon ligaments, the rotator cuff, and other supporting muscles [52]. It may be for this reason that, as the IR muscles fatigue due to low endurance, this process results in an increased load on the joint [52]. Similarly, this may explain why having a low strength ratio, likely due to a large difference between the strength between IR and ER movements, may result in an imbalance and predispose the joint to an injury.

### 4.1. Quality of Included Studies

The previously described quality scoring system presented in Table 2 displays that of the 13 included studies, only two [26,44] were scored ‘fair.’ The remaining 11 were scored as having ‘good’ methodological quality. The main area of concern within the methodological quality of included literature was in the justification of sample size [26,27,28,29,43,44,45,53] and, thus, articles not meeting an adequate sample size, as described in Section 2.8 Assessment of Sample Size, were excluded from the final review (Table 6). Other concerns in methodological quality of the included articles were primarily in the detail of the testing position [27,43,44] and confounding factors [48,49,51] (including not reporting the number of repetitions in testing).

### 4.2. Predictive Value of Isokinetic Dynamometry

Currently, there is only one previous systematic review [54] testing the predictive value of isokinetic dynamometry, which was done in the lower limb. The present study reveals strong evidence from prospective studies for isokinetic dynamometry measurements of IR endurance to predict in-season injury. This is inconsistent with Green et al.’s [54] systematic review on the lower limb, which found no predictive ability of isokinetic measurements. This shows low hamstrings and quadriceps’ strength and association with developing an injury. These inconsistencies may be attributable to the low sample sizes, difference in machine protocols, and difference in the joint or different demands of individual sports.

### 4.3. Strengths and Limitations

This review was limited by the availability of prospective literature and large sample sizes. Grey literature was excluded, which may increase the risk of publication bias [55]. Non-English studies were not included, which may have excluded high quality/relevant studies. The exclusion of handheld dynamometry may have influenced the publication bias. However, the aim of the study was made specific to isokinetic dynamometry. Hand-held dynamometry was previously studied by Furness et al. [22] and did not measure endurance or through range movement, which is more specific to sport.

Another potential limitation is that only significant findings were reported. Consequently, insignificant findings were not broadly discussed when included in this review. A quality assessment was done using a modified AXIS and modified CASP in which both do not have scoring systems to grade study quality. Therefore, one was developed for this review [36]. This could result in quality differences between other studies/reviews but was deemed necessary for standardising results and preventing selector bias. This review included cross-sectional studies, which was necessary due the limited amount of prospective literature on this topic. However, the results were split by study type with prospective literature being of higher quality. Data has been pooled based on study type to differentiate between the quality of evidence and to separate an association with injury from a potential predictor of injury. The inclusion of cross-sectional literature could be considered a limitation with regard to the aim of this study because they are of lower methodological quality and do not provide predictive ability. Looking at already injured athletes makes it difficult to ascertain if the cause of lower outputs are due to having pain or injury prior to initial testing. Given the higher level of evidence, results of the prospective studies were more highly considered in this review.

It also needs to be noted that, while cohort studies are used to determine predictors/risk factors of an injury, the exact mechanism is unable to be determined. Within this literature review, studies were selected where strength measures were conducted prior to a shoulder injury. The exact mechanism that resulted in the shoulder injury was directly related to the specific sport or training for that sport. For example, a volleyball athlete with a low IR isokinetic measure injures their shoulder by spiking the ball. In this case, the low IR isokinetic measure is the risk factor or predictor, while spiking the ball is the mechanism of injury.

Furthermore, this review included a variety of included sports reducing the review’s external validity to a single sport. An additional limitation is the confounding variables specific to training, playing, and conditioning that participants were exposed to within the included individual studies. Readers of this paper should be mindful of this limitation when generalising the results beyond this review.

An additional limitation to both study inclusion and comparison is the variability in the methods in which IR and ER were assessed. For example, the seat setting, shoulder positioning, and movements (eccentric and/or concentric) isokinetic speeds were not always specified or varied between studies. The nature of the variability in the included studies (sports) dictated the type of analysis but did not limit the ability to compare and synthesize the data. Whereby a meta-analysis was not possible, a best evidence synthesis was possible to provide recommendations on the strength of evidence.

Researchers were not able to conduct a meta-analysis due to heterogeneity of the included studies. The studies used a variety of measurements, settings, speeds, positioning, and warm-up protocols, which could affect outcomes and may be the reason why conflicting results were found. Researchers recommend future studies to follow uniform procedures and set-ups. Studies should also be more detailed in reporting their methods, including following a uniform procedure and the addition of consistently reporting chair angle and repetitions. This literature review has affirmed the need for more high-quality research to assess the predictability of isokinetic dynamometry in sports.

Strengths of this systematic review include the systematic approach employed and the rigorous methodology followed, using the PRISMA statement [31]. Additionally, two independent reviewers utilized the CASP and AXIS appraisal tools, which followed a high-quality assessment of methodological quality.

### 4.4. Implications for Future Research

The findings of this study can assist in determining what factors need to be accounted for when using isokinetic dynamometry testing to ensure high quality test-retest reliability. Baseline testing of individual sports will provide profiles of players/cohorts and ultimately determine thresholds for injury. At this stage, more research is needed in individual sporting populations using prospective cohort designs.

## 5. Conclusions

In conclusion, this review found a limited number of studies surrounding isokinetic dynamometry measurements of shoulder IR and ER in association with injury in overhead athletes specific to the research aim and most with limited sample sizes. When looking at the prospective studies, endurance scores and strength ratios were predictive of injury. Due to the conflicting results found in this review, more research in this area is needed to clarify findings. Notably, in the available prospective literature, isokinetic dynamometry measurements of IR endurance and strength ratios showed strong predictive ability for injury. Based on the paucity in high quality prospective studies, further research should place a focus on standardization of participant seat positioning (back rest, pan, and rotation angles), further specification on rest intervals, and predetermined outcome measures that would serve to improve the rigour of future studies. Future research should focus on establishing norms for each sport with a large sample size and conducting prospective studies to determine if isokinetic dynamometry measurements of shoulder rotation do predict injury in upper limb dominant sports.

## Figures and Tables

**Figure 1 sports-08-00124-f001:**
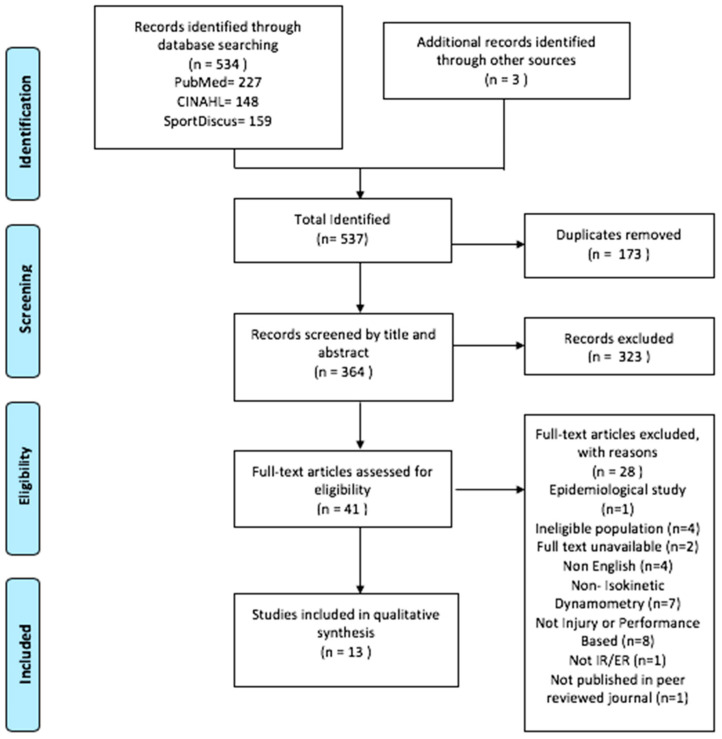
PRISMA flow diagram showing the literature search, screening, and eligibility results.

**Table 1 sports-08-00124-t001:** Search strategy used for the database search.

Database	Search Terms
PubMed, CINAHL & Sport Discus	(Biodex OR Cybex OR isokinetic* OR isotonic OR concentric OR eccentric OR “peak torque” OR dynamomet*) AND (Shoulder* OR Glenohumeral OR “Rotator Cuff” OR “rotator muscles” OR “upper limb” OR “upper extremity”) AND (swim* OR water-based OR surf* OR kayak* OR “water polo” OR “water sport” OR Baseball OR softball OR cricket OR volleyball OR “arm wrestling” OR sport OR sports OR sporting OR athlete* Or player* OR handball or rugby or basketball) AND (Injury OR injuries OR Strength OR “risk factors” OR preseason OR “weakness”) AND (Prospective OR prediction OR prevention OR predictor OR risk)

**Table 2 sports-08-00124-t002:** Critical appraisal results for all included studies.

Study (n=13) Author (Year) [Reference]	Scores Assigned by Item Number																		
AXIS	1	2	3	4	5	6	8	9	10	11	12	15	16	17	18	19	20	Score	Quality
Mickevicius et al. (2016) [27]	1	1	0	1	1	1	1	0	0	1	1	1	1	0	1	1	1	76%	Good
Stickley et al. (2008) [28]	1	1	0	1	1	0	1	1	1	1	1	1	1	0	1	1	1	82%	Good
Tonin et al. (2013) [43]	1	1	0	1	1	1	1	0	1	1	1	1	1	1	1	1	1	88%	Good
Bak et al. (1997) [44]	1	1	0	1	1	0	1	0	1	1	0	1	1	1	0	1	0	64%	Fair
Aginsky et al. (2004) [26]	1	1	0	1	0	0	1	1	1	0	1	1	1	0	0	1	0	58%	Fair
Beach et al. (1992) [45]	1	1	0	1	0	0	1	1	0	1	1	1	1	1	1	1	0	70%	Good
Stuelcken et al. (2008) [29]	1	1	0	1	1	0	1	1	0	1	1	1	1	1	1	1	1	82%	Good
CASP	1	2	3	4	5a	5b	6a	6b	7,8,9	10	11	12	Score	Quality					
Forthomme et al. (2018) [46]	1	1	1	1	1	1	1	1	1	1	1	1	100%	Good					
Forthomme et al. (2013) [47]	1	1	1	1	1	1	1	1	1	1	1	1	100%	Good					
McDonough et al. (2016) [48]	1	1	1	1	0	0	1	1	0	1	1	1	75%	Good					
Vogelpohl et al. (2015) [49]	1	1	1	1	0	0	1	1	1	1	0	1	83%	Good					
Edouard et al. (2013) [50]	1	1	1	1	1	1	1	1	1	1	1	1	91%	Good					
Wang et al. (2001) [51]	1	1	1	1	1	0	1	0	0	0	1	0	58%	Fair					

**Table 3 sports-08-00124-t003:** Key information from each included study organized by level of evidence.

Author (Year) and [Reference]	Aim/Objective/Hypothesis	Study Design	CASP/AXIS Score	Level of Evidence
Forthomme et al. (2018) [46]	To analyse measurements of maximal rotator muscle strength to identify intrinsic risk factors that could put elite handball players at risk for traumatic and micro-traumatic dominant-shoulder injuries.	Prospective Cohort	CASP, 100%	II
Forthomme et al. (2013) [47]	To highlight the intrinsic factors that could potentially put volleyball players at risk for shoulder injury, such as rotator cuff maximal strength, passive glenohumeral mobility, posterior rotator cuff stiffness, scapular resting position, or a forward presenting shoulder.	Prospective Cohort	CASP, 100%	II
Edouard et al. (2013) [50]	To analyse whether internal and external rotator shoulder muscles weakness and/or imbalance collected through a pre-season assessment could be predictors of subsequent shoulder injury during a season in handball players.	Prospective Cohort	CASP, 91%	II
Vogelpohl et al. (2015) [49]	To investigate the link between preseason shoulder rotator cuff functional strength ratios and the development of shoulder pain and injury.	Prospective Cohort	CASP, 83%	II
McDonough et al. (2014) [48]	To associate shoulder isokinetic strength and range of motion variable with subsequent injuries over a rugby league season.	Prospective Cohort	CASP, 75%	II
Wang et al. (2001) [51]	To evaluate the relationship between shoulder mobility, rotator muscles strength and scapular symmetry, and shoulder injuries and/or pain in elite volleyball athletes.	Prospective Cohort	CASP, 58%	II
Aginsky et al. (2004) [26]	To investigate the relationship between shoulder flexibility and isokinetic strength as possible factors that may predispose provincial fast bowlers to shoulder injuries.	Cross-Sectional	AXIS, 58%	III-3
Beach et al. (1992) [45]	To provide normative data on shoulder flexibility in swimmers, to determine if a correlation exists between flexibility and shoulder pain, and to determine the correlation between strength and endurance ratios to shoulder pain.	Cross-Sectional	AXIS, 70%	III-3
Bak et al. (1997) [44]	To examine shoulder strength and range of motion in two matched groups of swimmers with and without shoulder pain.	Cross-Sectional	AXIS, 64%	III-3
Mickevicius et al. (2016) [27]	To assess whether side-to-side differences in morphology and function of the upper limbs in 11–12-year-old male baseball players with throwing-related pain were more pronounced than that of age-matched healthy untrained subjects.	Cross-Sectional	AXIS, 76%	III-3
Stickley et al. (2008) [28]	To compare medial and lateral isokinetic peak torque of the rotator cuff among skill levels and between athletes with and without a history of shoulder injury.	Cross-Sectional	AXIS, 82%	III-3
Stuelcken et al. (2008) [29]	To determine the prevalence of shoulder pain in female cricket fast bowlers and compare the shoulder rotation range of motion and strength of those bowlers with and without a history of shoulder pain.	Cross-Sectional	AXIS, 82%	III-3
Tonin et al. (2013) [43]	To evaluate adaptive changes in the dominant shoulders of female professional overhead athletes, their mutual association, and relation between adaptive changes and shoulder injury.	Cross-Sectional	AXIS, 88%	III-3

**Table 4 sports-08-00124-t004:** Key characteristics and findings for each included study.

Author	Participant Details	Testing Protocol	Key Findings	
			Key Points	Measurements of Significant Findings
Aginsky et al. (2004) [26]	n = 21 M R arm fast bowlers A(y): 17–36, $\bar{x}=22$ (I) = 9 (C) = 12	Cybex -GHJ: 90° ABD -Elbow: 90° flexion -ROM: 150° Reps not specified, 90°/s, 180°/s	Strength: (I) had ↑ weight normalised Conc IR PT at 180°/s compared to (C) * -No significant difference between (I) and (C) groups for absolute PT at 90°/s or 180°/s for both IR & ER Ratio: No significant difference between (I) and (C) groups for ER:IR	PT at 180°/s -(I): $\bar{x}=65.20$ Nm/kg (10.03) -(C): $\bar{x}=45.91$ Nm/kg (10.26) *p* = 0.009
Beach et al. (1992) [49]	n = 32 (8 M, 24 F) Division 1 swimmers, and 4 club swimmers A(y): $\bar{x}=15–21$	Cybex II -Prone -GHJ: 90° ABD, Elbow: 90° flexion -3 reps of maximum effort at 60°/s, 50 reps at 240°/s	Ratio: Significant correlation to shoulder pain at 240°/s for ER * -Correlation to shoulder pain: L 0.61 R 0.69 *p* < 0.001 -Very low and nonsignificant correlation between strength ratios and shoulder pain	PT at 240°/s L: $\bar{x}=$ 80% (23%) R: $\bar{x}=$ 78% (22%)
Bak et al. (1997) [50]	n = 15 (6F, 9M) National level swimmers A(y): $\bar{x}=18–19$ (I) = 7 (C) = 8	Kin Com -Seated -GHJ: 80° ABD, 20° forward flexion in transverse plane -Elbow: 90° flexion 30°/s	Strength: No significant difference in PT for ER between (I) side compared to (C) side -No significant difference in PT for ER and IR btw (I) group & (C) group R: Fx ER Ecc: IR Conc at 30°/sec ↑ on (I) side compared to (C) side * -(I) group had significant ↑ Conc and Ecc ER:IR compared to (C) group *	FX ER Ecc:IR Conc -(I) side: $\bar{x}=$ 108% (18%) -(C) side: $\bar{x}=$ 89% (15%) ER:IR Conc and Ecc –(I) group: $\bar{x}=$ 83% (11%) -(C) group: $\bar{x}=$ 66% (11%) *p* = 0.02
Edouard et al. (2013) [43]	n = 30 F Elite handball players (16) and non-athletes (14) A(y): “youth” $\bar{x}=18$	Con Trex -Seated -GHJ: 45° ABD in scapular plane -Elbow: 90° flexion -ROM: GHJ, 70°, IR; 15°, ER; 55° -Conc; 3 reps 60°/s, 3 reps 120°/s, 5 reps 240°/s Ecc; 3 reps 60°/s	Strength: No significant relative risk of injury for Conc and Ecc at 60°/s & 120°/s Endurance: No significant relative risk of injury for Conc at 240°/s R: Relative risk of injury was 2.08 for Fx IR Ecc: ER Conc at 60°/s * -Relative risk of injury was 2.57 for conventional ER Conc: IR Conc at 240°/s *	ER Conc:IR Conc at 240°/s -criteria<.69 CI: 1.6–3.54, 95%, *p* < 0.05 Fx IR Ecc:ER Conc at 60°/s -criteria>1.61 CI: 1.18–2.98; 95%, *p* < 0.05
Forthomme et al. (2018) [44]	n = 108 M Handball, senior division A(y): $\bar{x}=24$ (I) = 51 (C) = 57	Cybex -Supine -GHJ: 90° ABD in frontal plane -Elbow: 90° flexion -ROM: 50° IR to 70° ER -Conc: 3 reps 60°/s, 5 reps 240°/s -Ecc: 4 reps 60°/s	Strength: No significant difference between (I) group and (C) group PT (*p* > 0.05) Endurance: (I) group had ↓ Conc IR at 240°/s compared to (C) group -Calculated odds ratio showed ↑ Conc IR at 240°/s was a protective factor * Ratio: No sig. diff for Conc ER:IR at 60°/s & 240°/s btw (I) & (C) groups	↑ conc IR at 240°/s -odds ratio = $\bar{x}=$ 0.93 (95%) CI = 0.865, 1.000, *p* = 0.49
Forthomme et al. (2013) [45]	n = 66 (34 M and 32 F) Competitive volleyball players A(y): $\bar{x}=24$ (I) = 15 (C) = 51	Cybex -Supine -GHJ: 90° ABD in frontal plane -Elbow: 90° flexion -ROM: 50° IR to 70° ER -Conc: 3 reps 60°/s, 5 reps 240°/s -Ecc: 4 reps 60°/s	Strength: (C) group had ↑ Ecc ER and IR at 60°/s * -Odds ratio showed Ecc contraction of IR and ER was a protective factor (odds ratio <1), and each ↑ of 1 N.m by IR & ER in the Ecc mode ↓ the risk of shoulder pain by 1% (Respective odds ratios = 0.946, *p* = 0.01 and 0.940, *p* = 0.05) * Endurance: No significant difference in IR and ER at 240°/s for both (C) & (I) Ratio: No significant difference in ER:IR for both (C) and (I)	Ecc ER & IR at 60°/s -IR: Ecc 60/s, (C): $\bar{x}=$ 51.2 N.m (17.4) (I): $\bar{x}=$ 38 N.m (12.3) *p* < 0.01 * -ER: Ecc 60/s; (C): $\bar{x}=$ 41.7 N.m (11) (I): $\bar{x}=$ 35.2 N.m (8.6) *p* < 0.05
McDonough et al. (2014) [47]	n = 20 M Professional/Semi-Professional Rugby League A(y): $\bar{x}=19$ (I) = 8 (C) = 12	Biodex -Seated -GHJ: 90° ABD in frontal plane -5 reps-180°/s	Strength: No significant difference in strength diff btw (I) and (C) groups -No predictive value for future injury Large effect for Ecc IR Ratio: No sig diff in IR Ecc:ER Conc	
Mickevicius et al. (2016) [27]	n = 30 M Baseball players A(y): 11–12 (I) = 14 baseball players with pain (C) = 16 age matched, healthy, non-athletes	Biodex -Seated -GHJ: 45° ABD, 30° horizontal flexion -Elbow: 90° flexion -ROM: 90° of extension to 180° flexion -3 maximal reps 90°/s	Strength: Ecc ER was ↑ in (C) group *p* < 0.05 * Ratio: ER Ecc: IR Conc was ↑ in (C) group (*p* < 0.05) *	Ecc ER (I) group: -Do: $\bar{x}=$ 16.8 N.m (5.6) -Ndo: $\bar{x}=$ 15.3 N.m (3.8) (C) group: -Do; $\bar{x}=$ 19.9 N.m (3.8) -Ndo: $\bar{x}=$ 20.3 N.m (6.6) ER Ecc:IR Conc (I): $\bar{x}=$ 55% (5%) (C): $\bar{x}=$ 64% (1%)
Stickley et al. (2008) [28]	n = 38 F Competitive volleyball players A(y): 10–15, $\bar{x}=13$ (I) = 9 (C) = 29	Biodex -Seated -GHJ: 30° flexion, 30° ABD -Elbow: 90° flexion -2 sets of 5 maximal repetitions 60°/s, 1st set Conc-2nd set Ecc	Strength: No significant difference between (I) and (C) groups Ratio: (I) group had ↓ IR Ecc: ER Conc compared to (C) group *	IR Ecc:ER Conc (I): $\bar{x}=$ 177% (39%) (C): $\bar{x}=$ 216% (44%) *p* = 0.02
Stuelcken et al. (2008) [29]	n = 26 F Elite fast bowlers A(y): $\bar{x}=22.5$ (I) = 12 (C) = 14	Kin Com -Seated -GHJ: 45° ABD, 30° horizontal flexion -Elbow: 90° flexion -90°/s 1 set 5 reps Conc > Ecc cycles	Strength: No bilateral diff in PT for (I) or (C) groups (*p* > 0.05). Ratio: No significant difference in ER:IR (*p* > 0.05)	
Tonin et al. (2013) [51]	n = 36 F Competitive Volleyball (15) and Handball (21) Unknown Age (I) = 14 (C) = 22	Biodex -Seated -GHJ: 90° ABD in scapular plane -Elbow: 90° flexion -ROM: 50° ER to 50° IR -20 maximal con reps 60°/s (endurance) -4 maximal reps 60°/s and 150°/s	Strength: (I) group had ↓ ER PT * -(I) group had ↓ Ecc ER PT at 60°/s * Endurance: (I) group had ↑ fatigability of IR and ER * Ratio: No significant difference for Ecc IR:ER Conc at 60°/s	ER PT (I): $\bar{x}=$ 120 (6.5), *p* = 0.021 (C): $\bar{x}=$ 129 (13.8) Ecc ER PT deficit at 60°/s (I): $\bar{x}=$ 14 (16.7), *p* = 0.049 (C): $\bar{x}=$ 2.8 (10.9) Fatigability (I): IR, $\bar{x}=$ 22.1% (10.4) ER; $\bar{x}=$ 28% (10.6) *p* = 0.013 (C): IR, $\bar{x}=$ 10.8% (20.5) ER; $\bar{x}=$ 16.6% (20.2) *p* = 0.028
Vogelpohl et al. (2015) [48]	n = 15 M Collegiate Baseball Players A(y): $\bar{x}=9.5$ (I) = 6 (C) = 9	Biodex -Seated -GHJ: 45° ABD, 30° horizontal flexion -Conc and Ecc 4 reps 60°/s, 180°/s, and 300°/s 6 trials total	Strength: No sig. diff btw PT at 60°/s or 180°/s for (I) and (C) groups Endurance: Sig.↓ IR Conc PT at 300°/s in the (I) group (*p* = 0.003) * Ratio: Sig. ↑ ER Ecc:IR Conc (acceleration phase) at 300°/s in (I) group compared to (C) group. *	IR Conc PT at 300°/s (I): $\bar{x}=$ 34.73 N.m (13.71) (C): $\bar{x}=$ 55.82 N.m (8.06), *p* = 0.003 ER Ecc:IR Conc at 300°/s (I): $\bar{x}=$ 177% (107%), *p* = 0.02 (D): $\bar{x}=$ 81% (17%)
Wang et al. (2001) [46]	n = 16 M National level Volleyball players Unknown Age	Kin Com -Supine -GHJ: 90° ABD Elbow: 90° flexion -ROM: 50° ER to 50° IR -3 maximal contractions 60°/s, and 180°/s	Strength: No significant correlation between injury and muscle weakness Ratio: Significant correlation between muscle imbalance and injury (*p* = 0.041) Association between shoulder muscle strength imbalance in dominant arm and injury was statistically significant (*p* < 0.05) *	

$\bar{x}$, Mean; (C), Control Group (uninjured); (I), Injured Group (injured, symptomatic, painful); L, Left; R, Right; Fx, Functional; Ecc, Eccentric; Conc, Concentric; IR, Internal Rotation; ER, External Rotation; GHJ, Glenohumeral Joint (shoulder); ER:IR, External Rotation to Internal Rotation Ratio; PT, Peak Torque; Sig., Significant; Sig. Diff, Significant Difference; Diff, Difference; Btw, Between; A(y), Age in Years; n, Number; M, Male; F, Female; R, Right; L, Left; ROM, Range of Motion; ABD, Abduction; Do, Dominant Shoulder; Ndo, Non-dominant shoulder; °/s, Degrees per second; ↑, increased (higher, greater); ↓, decreased (lower); &, and; Corr., correlation; CI, Confidence interval; reps, Repetitions; *, significant finding (*p* < 0.05).

**Table 5 sports-08-00124-t005:** Effects for each outcome measured.

Outcome Measured	Effect (+/−/=) [Article Reference]	Best Evidence Synthesis *
Strength (PT)	(+) [27,43,46,47]	Conflicting
	(−) [26]	
	(=) [28,29,44,46,48,49,50,51]	
Endurance	(+) [43,46,49]	Conflicting
	(=) [47,50]	
Strength (PT) Ratio	(+) [27,28,50,51]	Conflicting
	(−) [44]	
	(=) [26,29,43,45,46,47,48]	
Endurance Ratio	(+) [45,50]	Conflicting
	(−) [49]	
	(=) [46,47]	

+, positive finding-in favour of control group, −, positive finding-in favour of injured group, =, no difference between groups. *** Conflicting evidence based on inconsistent findings in multiple studies (<75% of the studies reported consistent findings) [40].

**Table 6 sports-08-00124-t006:** Final summary of outcomes and effects.

Study	Outcome			
Author (Year) [Reference]	Strength	Endurance	S: Ratio	E: Ratio
Cross Sectional	+/−	+/−	+/−	+/−
Mickevicius et al. (2016) [27]	+		+	
Stickley et al. (2008) [28]			+	
Tonin et al. (2013) [43]	+	+		
Bak et al. (1997) [44]			−	
Aginsky et al. (2004) [26]	−			
Beach et al. (1992) [45]				+
Stuelcken et al. (2008) [29]				
Best Evidence Synthesis *	Conflicting	Limited	Conflicting	Limited
Prospective				
Forthomme et al. (2018) [46]		+		
Forthomme et al. (2013) [47]	+			
McDonough et al. (2016) [48]				
Vogelpohl et al. (2015) [49]		+		−
Edouard et al. (2013) [50]			+	+
Wang et al. (2001) [51]			+	
Best Evidence Synthesis *	Limited	Strong	Strong	Conflicting

+, positive finding-in favour of the control group, −, positive finding-in favour of the injured group, * Strong evidence based on two or more studies with high quality and by generally consistent findings in all studies. Moderate evidence based on one study with high quality and/or two or more studies with low quality, and by generally consistent findings in all studies (≥75%). Limited evidence based on only one study. Conflicting evidence based on inconsistent findings in multiple studies (<75% of the studies reported consistent findings) [39].

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
