# Peer review of "Isokinetic Dynamometry as a Tool to Predict Shoulder Injury in an Overhead Athlete Population: A Systematic Review"

_sports, 2020, doi:10.3390/sports8090124_

Round 1

Reviewer 1 Report

sports-849802

Isokinetic Dynamometry as a Tool to Predict Shoulder Injury in an Overhead Athlete Population: A Systematic Review

Thanks for the opportunity to review your paper. Your methods and systematic review process are clearly described. The discussion is relatively brief - the limitations section is particularly well done. My overall concern with the manuscript was a fairly overwhelming ‘so what?’. I’ve not kept up with the area, but I had assumed that interest in isokinetic ratios and normative data had dissipated in the late 1980s - clearly I was wrong. While I am impressed with the methodology I did think the manuscript needs a much stronger rationale and basis for discussing the results.

Reading your manuscript prompted a number of questions for me. Consideration of these may (in some instances) help improve your paper (or not).

  1. Shoulder injuries is a broad categorisation - really an anatomical region rather than specific injuries. The argument is made for the muscular strength and endurance of shoulder internal and external rotational movements - while these are no doubt important for certain injury etiologies they arguably have no relationship with others. It is not clear to me how distinctions are made in your systematic review. For example you have swimmers (relatively slow repetitive concentric actions), volleyballers and cricket bowlers (isolated high velocity actions with eccentric decelerations) and rugby league and handball ( multiple actions with high risk of impact injuries/collisions).
  2. No real rationale for IR:ER strength, endurance relationships is provided - I believe that readers should be offered some explanation/rationale for the study purpose and search criteria. The concentric acceleration and eccentric deceleration muscular actions also would benefit from discussion - some of your reviewed manuscripts have included data on these suggested functional deficits
  3. Given that the studies cited seemed to have assessed IR and ER in various ways, I wonder whether descriptors and discussions should refer only to these movements and avoid suggesting specific muscle groups - which arguably can not be isolated for these movements - e.g. internal and external rotator shoulder muscles
  4. There were notable differences in the studies for elements such as positioning for IR/ER assessment, isokinetic speeds utilised, and the mix of concentric and eccentric actions - these perhaps should be included with limitations.

Minor points/typographical questions:
L 39 ‘through fluctuating its resistance.’ - does it really - not my understanding of isokinetic dynamometry

l57/58 As the primary muscles used in throwing and swimming sports are the rotator cuff muscles - they undoubtedly play a role in control and joint stability, but they are certainly not prime movers for these kinds of actions

L 69 previously mentioned above. - one of those words seems redundant

L 99 google scholar ? Capitalisation

Best wishes moving forward with this.

Author Response

To:                

Prof. Eling Douwe De Bruin,

Editor, MDPI Sports

From:

Andrea Bagordo

Kimberly Ciletti 

 A/Prof Kevin Kemp-Smith
  Dr Vini Simas
  Dr Mike Climstein
  A/Prof James Furness

Date:            20th July 2020

Re:               Isokinetic Dynamometry as a Tool to Predict Shoulder Injury in an Overhead Athlete Population: A Systematic Review (Sports-849802)

______________________________________________________________

Reviewer’s comments

Author’s reply

Reviewer 1

1) Shoulder injuries is a broad categorisation - really an anatomical region rather than specific injuries. The argument is made for the muscular strength and endurance of shoulder internal and external rotational movements

While these are no doubt important for certain injury etiologies they arguably have no relationship with others. It is not clear to me how distinctions are made in your systematic review. For example you have swimmers (relatively slow repetitive concentric actions), volleyballers and cricket bowlers (isolated high velocity actions with eccentric decelerations) and rugby league and handball (multiple actions with high risk of impact injuries/collisions).

✓   We thank Reviewer 1 for his/her review of our manuscript.   We agree with Reviewer 1 that injuries can be broadly categorized by anatomical region however our previous studies and other research commonly reports anatomical region(s) (separate or combined), mechanism(s) of injury, location or classification (musculoskeletal, kin, joint, etc.) incidence and severity 1,2. We acknowledge the author’s comments around the diversity of sports and actions included in this review. The broad nature of this literature review was not limited to one sport but to all upper limb dominant sports as there was not enough literature available to conduct a more specific systematic review at this time.

Our inclusion of research findings with swimmers, volleyball, cricket bowlers and handball (all with varying limb velocities as noted by Reviewer 1) was to  broadly associate to surfing (this was a preliminary study for a follow up cross-sectional manuscript on surfers and shoulder isokinetic dynamometry also submitted to Sports and currently under review). Surfing is an upper body dominant aerobic/anaerobic activity with other similar activities which there is little research on.

As noted by this literature review, there is limited research available conducted on isokinetic dynamometry and its association to shoulder injury and even less when a specific sport is selected. Therefore there is inadequate literature currently available to complete a systematic review on surfing, swimming, or any individual upper limb dominant sport using isokinetic dynamometry in association to injury.

With regard to rugby league/handball and high risk of impact we argue that surfers do indeed experience impact, both from striking the seafloor or sea surface (18%), struck by their own board (36.7% - 73.4%, depending upon the data source, emergency department or survey data) or struct by another surfers board (~6.5%) 2.

1.     Furness et al., (2015).  Acute injuries in recreational and competitive surfers: incidence, severity, location, type, and mechanism.  Am J Sports Med. 43(5). 1246-1254.

2.     McArthur et al., (2020). Epidemiology of Acute Injuries in Surfing: Type, Location, Mechanism, Severity, and Incidence: A Systematic Review. Sports. 8(2):1-25.

2. No real rationale for IR:ER strength, endurance relationships is provided - I believe that readers should be offered some explanation/rationale for the study purpose and search criteria. The concentric acceleration and eccentric deceleration muscular actions also would benefit from discussion - some of your reviewed manuscripts have included data on these suggested functional deficits

✓   We agree with Reviewer 1’s comment with regard to expanding the rationale for our study and we have amended our manuscript accordingly.

We have provided further rationale on lines 89-93 stating,

The IR:ER strength, endurance relationship is further discussed in our discussion on lines 40-48;  however, it was not the purpose of this review to study the physiological reason for the development of injury but rather to find associative or predictive factors. As the studies were heterogeneous in their protocols (i.e. concentric and/or eccentric, angles and speeds) we cannot further develop the second sentence from the reviewer in this study and this has been included as a limitation and recommendation for future research to follow uniform procedures and set-ups.

 In brief, isokinetic evaluation of the shoulder is used to assess function stability and the muscular strength/endurance of the shoulder musculature. Lines  59-62 have been amended to clarify: The evaluation of the shoulder is often used to assess functional stability, dynamics and muscular performance (strength, speed and endurance) in sports that are predominantly overhead.  Additionally, isokinetic dynamometry evaluation has also been used to determine profiles or athletes and patients who demonstrate shoulder abnormalities and used in clinical decision making and rehabilitation.”

Noffal GJ (2003).  Isokinetic eccentric-to-concentric strength ratios of the shoulder rotator muscles in throwers and non-throwers. Am J Sports Medicine. 31: 537–541. 

•  Batalha et al., (2020).  The Effectiveness of a Dry-Land Shoulder Rotators Strength Training Program in Injury Prevention in Competitive Swimmers. J of Human Kinetics. 71:11-20. 

3) Given that the studies cited seemed to have assessed IR and ER in various ways, I wonder whether descriptors and discussions should refer only to these movements and avoid suggesting specific muscle groups - which arguably can not be isolated for these movements - e.g. internal and external rotator shoulder muscles

✓ We agree with Reviewer 1 and have made appropriate amendments to the manuscript (where all IR & ER muscles are referred to were changed to movement).

4)         There were notable differences in the studies for elements such as positioning for IR/ER assessment, isokinetic speeds utilised, and the mix of concentric and eccentric actions - these perhaps should be included with limitations.

✓   We agree with Reviewer 1 and have included these differences in the Limitations section of our manuscript.

An additional line has been added (lines 95-97), this now reads: “An additional limitation to both study inclusion and comparison is the variability in the methods in which IR and ER were assessed. For example, the seat setting, shoulder positioning, movements (eccentric and/or concentric) isokinetic speeds were not always specified or varied between studies.”

Minor Points/typographical questions: L 39 ‘through fluctuating its resistance.’ - does it really - not my understanding of isokinetic dynamometry

✓   We have amended this sentence to better and deleted the term “fluctuating its resistance”.

l57/58 As the primary muscles used in throwing and swimming sports are the rotator cuff muscles - they undoubtedly play a role in control and joint stability, but they are certainly not prime movers for these kinds of actions

✓   Our use of the term “primary” in this sentence does not refer to “primary movers”, we had amended this sentence and removed the word “primary” to avoid any confusion.

L 69 previously mentioned above. - one of those words seems redundant

✓   We have amended that sentence so the word “coach: does not appear twice.

L 99 google scholar ? Capitalisation

✓   We have corrected to Google Scholar

Reviewer 2 Report

SPORTS_849802

Isokinetic dynamometry as a tool to predict shoulder injury in an overhead athlete population: a systematic review

This study identified isokinetic dynamometry studies addressing external rotation (ER) or internal rotation (IR) strength and endurance in the overhead athletic population and its relation to athletic injury. A total of 13 articles were included in the study. The authors found that cross-sectional studies reported conflicting results while prospective study designs revealed strong evidence that reduced IR endurance and strength ratios are predictive of shoulder injury. It was concluded that more prospective studies are needed in individual sporting population.

GENERAL COMMENTS TO THE AUTHORS

This study presents an examination of the role of isokinetic dynamometry to predict sport injury. The findings of the study are of practical relevance for coaches and practitioners for determining training and preventive strategies.

SPECIFIC COMMENTS

Introduction

  1. Lines 34-41: I feel that these sentences are disconnected. Please, try to ‘connect’ the information presented.
  2. Lines 42-44: Please rewrite to improve clarity.
  3. Lines 51-52: Please rewrite to improve clarity.
  4. Please provide a better rationale for the study
  5.  

Methods

  1. Was the study protocol registered in PROSPERO? Please, clarify.
  2. Line 112: eligibility criteria were discussed. Please, correct.
  3. Line 254: rest a period: please correct to ‘resting period’.
  4. Line 298: please clarify which studies were disregarded.

Results

  1. Please modify Table 4 according to the journal Instructions for authors

Author Response

Reviewer 2

We thank Reviewer 2 for his/her review of our manuscript.

Introduction: Lines 34-41: I feel that these sentences are disconnected. Please, try to ‘connect’ the information presented.

✓   We have amended these lines to flow better

Lines 42-44: Please rewrite to improve clarity.

✓   We have amended these lines to flow better

Lines 51-52: Please rewrite to improve clarity.

✓   We have amended this sentence to improve clarity

Please provide a better rationale for the study

✓   We have amended our rationale for the study to include:

Following a thorough and extensive review of the literature, there is currently no systematic review specific to strength measurements of shoulder IR and ER using isokinetic dynamometry and IR, ER and IR : ER ratios and association with injury. Given the high rate of shoulder injury across a variety of overhead predominant sports, assessment of IR and ER via isokinetic dynamometry may provide valuable insight into injury prevention.”

Methods:  Was the study protocol registered in PROSPERO? Please, clarify.

✓   The protocol was developed prospectively prior to data extraction and critical appraisal; however, it was not registered within an online registry such as Prospero.

Added to lines 135-136 in methods

Line 112: eligibility criteria were discussed. Please, correct.

✓   We have amended the text as recommended by Reviewer 2

Line 254: rest a period: please correct to ‘resting period’.

✓   We have amended the text as recommended by Reviewer 2

Line 298: please clarify which studies were disregarded.

✓  We understand that the Reviewer is requesting the specific studies which did not meet significant findings with a low sample size. From 2.8 Assessment of Sample Size in the amended manuscript, we would like to clarify the following:

However, articles that did not detect a statistically significant relationship and had a sample size that did not meet requirements laid out by Hsieh et. al [37] were removed due to the having a sample size too small to truly detect differences between groups (Table 6).”

Hsieh is not an article included in our review (as outlined in the  PRISMA selection of studies flow diagram); it is the reference used to cite ‘adequate’ sample size criteria. 

i.e. only one article had over 100 participants, and any one study could have measured strength, endurance and ratios and only found significant findings in one of these outcome measures (hence an * (asterisk) sign in the data extraction table (T. 4) and inclusion in our final table (T.6) --- We did not include findings that did not show significance (P>0.05) AND had an inadequate sample size- (T. 6). 

There was a mistake in the table referenced attached to this comment (this has been changed from Table 5 to Table 6)

Table 6 compares the results after separating cross-sectional and prospective cohort studies, after the removal of non-significant findings in articles that did not reach the participant requirement outlined by Hsieh[37]. This simplistic view makes it easier to view the statistically significant evidence and the relationships between the studies.

Results:  Please modify Table 4 according to the journal Instructions for authors

✓ Table 4 has been changed from Landscape to Portrait

Reviewer 3 Report

Glad to have an opportunity to review this manuscript, but first of all, this manuscript cannot be accepted as its current form and format.

 There are severe problematic areas of the manuscript and the authors were not able to deal with the essential aspects of so-called “scientific research.”

This study cannot be progressed into any further steps of publication in this quality journal unless the following issues properly dealt with:

The abstract not successfully compiling and summarizing focal points of this study

The authors were not successfully provide adequate rationales for their comments in many of sentences especially when they commented about the selection of subjects and methodology part. It is too brief and doesn’t provide quality justification of the purpose statement.

Validity issue – your study is not successful to provide validity evidence of your measurement issues. I am not really convinced to your findings and not clear about what would be potential lessons from reading your study.

Please check and re-confirm and have other experienced scholars to read your manuscript prior to “submission” in terms of “research process” and conclusion part. Your current conclusion is still too brief and not really meaningful.

Finally authors, please should considerer to include some of the following references

[1]      C. Calvo-Lobo, S. Pacheco-da-Costa, J. Martínez-Martínez, D. Rodríguez-Sanz, P. Cuesta-Álvaro, D. López-López, Dry Needling on the Infraspinatus Latent and Active Myofascial Trigger Points in  Older Adults With Nonspecific Shoulder Pain: A Randomized Clinical Trial., J. Geriatr. Phys. Ther. 41 (2018) 1–13. https://doi.org/10.1519/JPT.0000000000000079.

[2]      C. Calvo-Lobo, S. Pacheco-da-Costa, E. Hita-Herranz, Efficacy of Deep Dry Needling on Latent Myofascial Trigger Points in Older Adults  With Nonspecific Shoulder Pain: A Randomized, Controlled Clinical Trial Pilot Study., J. Geriatr. Phys. Ther. 40 (2017) 63–73. https://doi.org/10.1519/JPT.0000000000000048.

[3]        C. Calvo Lobo, C. Romero Morales, D. Rodríguez Sanz, I. Sanz Corbalán, E.A. Sánchez Romero, J. Fernández Carnero, D. López López, Comparison of hand grip strength and upper limb pressure pain threshold between  older adults with or without non-specific shoulder pain., PeerJ. 5 (2017) e2995. https://doi.org/10.7717/peerj.2995.

Author Response

Reviewer 3

The abstract not successfully compiling and summarizing focal points of this study

✓   We wish to thank Reviewer 3 for their review of our manuscript.

The authors were not successfully provide adequate rationales for their comments in many of sentences especially when they commented about the selection of subjects and methodology part. It is too brief and doesn’t provide quality justification of the purpose statement.

✓   We have amended the manuscript to improve the rationale for our study.

Validity issue – your study is not successful to provide validity evidence of your measurement issues. I am not really convinced to your findings and not clear about what would be potential lessons from reading your study.

Justification:  it was not the purpose of our study to validate Isokinetic Dynamometry, rather to complete a systematic review on its use with regard to athletes who are involved in a predominantly overhead sport. Authors used a systematic approach and rigorous methodology using the PRISMA statement guideline. Additionally, two independent reviewers utilized the CASP and AXIS appraisal tools which followed a high-quality assessment of methodological quality.

 Our findings indicate that low isokinetic endurance of the shoulders (IR) and low strength ratios were predictors of injury, which will assist strength and conditioning coaches and rehabilitation specialists in both prevention and rehabilitation.

The findings of this study can assist in determining what factors need to be accounted for when using isokinetic dynamometry testing to ensure high-quality test-retest reliability and we have shown that high-quality prospective research shows strong evidence that upper limb dominant athletes with lower endurance measurements for internal rotation and low strength ratios IR:ER developed a shoulder injury in their sporting season.

Please check and re-confirm and have other experienced scholars to read your manuscript prior to “submission” in terms of “research process” and conclusion part. Your current conclusion is still too brief and not really meaningful.

Justification:  the senior authors are widely published and can assure Reviewer 3 that the manuscript was thoroughly read and appropriate edits made before submission.   We have however, re-read the manuscript and made subtle edits throughout.

Finally authors, please should considerer to include some of the following references

[1]      C. Calvo-Lobo, S. Pacheco-da-Costa, J. Martínez-Martínez, D. Rodríguez-Sanz, P. Cuesta-Álvaro, D. López-López, Dry Needling on the Infraspinatus Latent and Active Myofascial Trigger Points in  Older Adults With Nonspecific Shoulder Pain: A Randomized Clinical Trial., J. Geriatr. Phys. Ther. 41 (2018) 1–13. https://doi.org/10.1519/JPT.0000000000000079.

[2]      C. Calvo-Lobo, S. Pacheco-da-Costa, E. Hita-Herranz, Efficacy of Deep Dry Needling on Latent Myofascial Trigger Points in Older Adults  With Nonspecific Shoulder Pain: A Randomized, Controlled Clinical Trial Pilot Study., J. Geriatr. Phys. Ther. 40 (2017) 63–73. https://doi.org/10.1519/JPT.0000000000000048.

[3]        C. Calvo Lobo, C. Romero Morales, D. Rodríguez Sanz, I. Sanz Corbalán, E.A. Sánchez Romero, J. Fernández Carnero, D. López López, Comparison of hand grip strength and upper limb pressure pain threshold between  older adults with or without non-specific shoulder pain., PeerJ. 5 (2017) e2995. https://doi.org/10.7717/peerj.2995.

Justification:  Reviewer 3 has recommended the inclusion of three references. Unfortunately, after looking into these references, the first two studies are not related to our present study/manuscript and none of which meet our inclusion criteria.

Please refer to our inclusion criteria: on lines 155- 167, specifically, Studies included needed to utilise isokinetic dynamometry as an intervention and be either observational design (prospective or retrospective cohort study or cross-sectional study) to ensure all existing literature meeting the aim would be reviewed. Additionally, studies needed to examine shoulder strength and/or endurance, include shoulder IR and ER as a measurement outcome and either analyse the risk of injury (prospective or retrospective cohort) or compare uninjured versus uninjured participants (cross-sectional). Any study not meeting the inclusion criteria or that only profile strength values or injuries in a sporting population and did not link outcomes to injury were excluded.”

However, our current paper in development which is investigating IR and ER strength profiles in surfers with a current (or past) shoulder injury is a good fit for the Calvo-Lobo et al., (2017) paper (PeerJ) and we thank Reviewer 3 for bringing this publication to our attention as we will incorporate it into our manuscript in progress.

Reviewer 4 Report

It is difficult to provide specific comments and review because there are no line determinations.

Introduction, first paragraph- Isokinetics may be the gold standard for measurement through a range of movement, but has it been shown that that specific range of motion is related to actual sports functional movements. It is always the same problem with isokinetics- is it physiologic or related to functional physiology. Until that question is better answered in this paper, the underlying reason for studying isokinetics is not clear.

Introduction, fourth paragraph- rugby and surfing, by the author's definition, are not overhead sports. They have high rates of upper extremity injury, but the mechanics are no overhead. They should not be used to discuss issues of overhead sports motions.

3.3.2.- The wide variability in the sports makes conclusions difficult. The mechanics of each of the sports studied are very different. The playing conditions are different, and there is no control over any conditioning or exercises that may influence the strength or endurance

3.3.3 Testing protocol- this is a major limitation in the study. With the small number of studies, variations in how the data is collected become large counfounding problems. The variability in testing positions, arm positions, and testing speeds means that the data is not comparable, unless more testing can demonstrate any correlations.

Limitations third paragraph- This is a major limitation and probably makes  the data not comparable because of the wide variability. Another limitation regarding the isokinetic testing is whether any position is best or if the testing speed or the testing position has any relation to the actual physiological demands of any overhead sport. Other limitations include variability of the sports evaluated, and no definition of pain or injury.

Implications- The major implication from this analysis is how difficult it is to study injury in the multifaceted overhead motion. The discussion may point out the major difficulties and point to ways to improve the process.

Reviewer 5 Report

This is a nice paper and job well done. My specific remarks are written below. My major concern is the inclusion of cross sectional studies. You have written yourself in the discussion the problem with such studies, so I am not sure why didn't you exclude them straight away. Otherwise, great paper, interesting topic, filling the missing gap in knowledge. 

“The incidence of shoulder injuries in the overhead sporting population has been described in the literature as 0.2/1000 hours and 1.8/1000 hours [18-20]. “ Could you also provide some prevalence data for those sports, as overuse is actually main issue with shoulder

“None of the included studies provided information on the specific chair angle used” You are probably referring to chair backrest. Please clarify this, as on the majority of the dynamometers you can adjust the chair angle as well, as chairs can rotate.

You should clarify what does endurance (as outcome) means. Same goes for endurance ratio.  Is it fatigue index or ratio at medium/high speed? This should be clear to the reader. I consider myself an expert in the field of isokinetic, but Table 4 is hard to follow. Too many information put in single table. I don’t believe that layman reader would find such table useful.

“Of these, two good quality articles [27,51] found an association between a lack of strength and injury and one…” You are talking about cross sectional studies. How can there be an association when study design for analysing association and risk is inappropriate. I believe that only prospective studies should be included, otherwise you can not claim whether the finding is a cause or consequence of the injury.  This is also my major concern and remark regarding your paper. 

In prospective studies you have 2 studies that did were not properly powered (low number of participants). Why didn’t you exclude those studies in the first place?

I would add something in the conclusion as a guideline on how should future studies be conducted (velocity, testing position, rest period, outcome measures) with proposed statistical approach (if looking for cutoff, then ROC analysis would be in place).  

Round 2

Reviewer 1 Report

Thank you for the revisions and your consideration of comments from reviewers. I appreciate the changes which have, I think, improved your manuscript, however my original assessment has not been modified. As a stand alone manuscript - which I have to review this as - I cannot get past the absence of links between injury types, possible mechanisms and isokinetic measures. In my reading of your manuscript there are injuries included in the data sets which have no apparent links to either muscular strength or muscular endurance. Overall this is a length manuscript which does not really offer any usable take home messages or advance our understanding of injury prevention. Best wishes 

Reviewer 3 Report

This study supports novel information about Isokinetic Dynamometry as a Tool to Predict Shoulder Injury in an Overhead Athlete Population

This is an interesting aim with the quality of life scope. 

Authors have adressed all the required modifications in a correct way

The redaction is clear and concise with appropriated scientific terms.

The sample size calculation, structured tables and methodology are adequate and provide important contents.

Therefore, this study may support considerations about Isokinetic Dynamometry as a Tool to Predict Shoulder Injury in an Overhead Athlete Population

Author Response

Thank you for your review and passing our review for publishing. 

Reviewer 4 Report

The authors adequately addressed the concerns raised in the original review